# Cloning and Functional Analysis of *NtMYB9* in 'Jinzhanyintai' of *Narcissus tazetta* var. *chinensis*

Keke Fan [1,2,†], Yanjun Ma [1,2,†], Yanting Chang [1,2], Xiaomeng Hu [1,2], Wenbo Zhang [1,2], Yayun Deng [1,2], Mengsi Xia [1,2], Zehui Jiang [1,2,*] and Tao Hu [1,2,*]

1    International Center for Bamboo and Rattan, No. 8, Futong Eastern Avenue, Wangjing Area, Chaoyang District, Beijing 100102, China; fankk@icbr.ac.cn (K.F.); mayanjun@icbr.ac.cn (Y.M.); changyanting@icbr.ac.cn (Y.C.); xm766408356@126.com (X.H.); wenbozhang@icbr.ac.cn (W.Z.); yayundeng@icbr.ac.cn (Y.D.); xiamengsi@icbr.ac.cn (M.X.)
2    Key Laboratory of National Forestry and Grassland Administration/Beijing for Bamboo & Rattan Science and Technology, No. 8, Futong Eastern Avenue, Wangjing Area, Chaoyang District, Beijing 100102, China
*    Correspondence: jiangzh@icbr.ac.cn (Z.J.); hutao@icbr.ac.cn (T.H.)
†    These authors contributed equally to this work.

**Abstract:** *Narcissus tazetta* var. *chinensis* M.Roem. (Chinese Narcissus) is a traditional and famous flower in China, and its corona shows a transition from green to yellow with the opening of flowers, which is of great ornamental value. To investigate the molecular mechanism of the corona color-forming in the Chinese Narcissus, in this study, a MYB gene was screened from the transcriptome of the corona by bioinformatics analysis methods, named *NtMYB9*. The expression level of the *NtMYB9* gene was significantly higher in the corona than in the tepal, filament, ovary and leaf, and reached the highest level at the fourth period of complete coloration of the corona. The full-length sequence of the *NtMYB9* coding region was cloned using the corona cDNA as a template. Sequence analysis revealed that NtMYB9 protein contained R2 and R3 domains, phylogenetic tree analysis showed that the NtMYB9 protein was most closely related to flavonol activators. Subcellular localization showed that NtMYB9 was localized in the nucleus. The overexpression of *NtMYB9* gene into tobacco leaves and the expression level of seven enzyme genes related to the flavonoid biosynthesis pathway were significantly up-regulated. The transient transformation of *Petunia × hybrida* (Petunia) and *Phalaenopsis aphrodite* (Phalaenopsis) significantly revealed that the *NtMYB9* gene could turn flowers yellow. These results suggest that the *NtMYB9* gene is involved in the positive regulation of flavonoid biosynthesis in the Chinese Narcissus and may promote the synthesis of flavonols. In conclusion, these findings provide a valuable resource for further studies on the regulatory mechanism of the flavonoid biosynthesis pathway, and they are also beneficial to the molecular breeding of Chinese Narcissus.

**Keywords:** Chinese Narcissus 'Jinzhanyintai'; corona; flavonoid biosynthetic pathway; R2R3-MYB; transcriptional activator

## 1. Introduction

Chinese Narcissus (*N. tazetta* var. *chinensis* M.Roem.) is a traditional and famous flower in China [1], with beautiful posture, bright flower color, beautiful flower shape, freshness and elegance, which is of great ornamental value [2]. Compared with other flowers, the Chinese Narcissus lack the key gene *ANS* for anthocyanin synthesis, so there are fewer color varieties, and the flower colors are mainly yellow and white. Flower color is one of the most important traits of ornamental plants, and 'Jinzhanyintai', as a cultivar of Chinese Narcissus, has six white tepals and a yellow wineglass-shaped corona. Yellow is popular as the symbolic color of Chinese culture and Chinese civilization, and the wineglass-shaped corona is of great ornamental value [3,4]. It is one of the hot spots for breeders to analyze the mechanism of flower color-forming and to innovate new varieties of flower color using molecular biology tools.

The research on the flavonoid biosynthetic pathway of plant is relatively mature, and the metabolic pathway has been basically clarified, in which the flavonoid metabolites are one of the most important pigments of ornamental plants [5]. In higher plants, the flavonoid biosynthesis pathway is influenced by a variety of transcription factors, and the transcription factors in the flavonoid synthesis pathway are mainly MYB, bHLH and WD40, etc. The MBW (MYB-bHLH-WD40) complex is also jointly involved in the expression of related enzyme genes in the flavonoid synthesis pathway [6]. The MYB is one of the transcription factor families with the largest number of studies and the most extensive roles in ornamental plants. Most MYB transcription factors belong to R2R3-MYB [7], which directly regulates the transcription and expression of related enzyme genes by binding to their promoters [8], thereby activating or repressing the synthesis of flavonoid metabolites [9], which in turn affects the production and accumulation of phytochrome [10]. Currently, many MYB transcription factors that regulate plant flower color have been identified in many ornamentals and model plants. Some of these MYB factors can activate the expression of relevant genes in the flavonoid biosynthetic pathway and promote the synthesis and accumulation of anthocyanins [11], such as AtMYB75 and AtMYB90 in *Arabidopsis thaliana* [12], MdMYB1 in *Malus domestica* [13], and VvMYBA1 and VvMYBA2 in *Vitis vinifera* [14,15]. Several MYB transcriptional repressors have also been identified in the flavonoid biosynthesis pathway of plants, such as AtMYB3 and AtMYB4 in *A. thaliana* [16], FaMYB1 in *Fragaria × ananassa* [17], PpMYB18 in *Prunus persica* [18], VvMYBC2-L2 and VvMYBC2-L3 in *V. vinifera* [19,20], which inhibit the expression of downstream genes related to the flavonoid biosynthesis pathway, thereby suppressing the synthesis of anthocyanins.

In recent years, R2R3-MYB transcription factors have been characterized to regulate the flavonoid synthesis pathway in the corona of the Chinese Narcissus, but most of them are transcriptional repressors [21], and there are fewer studies about the flavonoid biosynthesis pathway in the color-forming mechanism of the corona during the flowering of the Chinese Narcissus. Therefore, the research on the color-forming mechanism of the corona is very meaningful.

In this study, based on the transcriptomic data of the corona coloring process in the Chinese Narcissus 'Jinzhanyintai' (unpublished), using the MYB protein sequences of *A. thaliana* and *O. sativa* as query sequences and identifying the MYB sequences of the Chinese Narcissus by HEMMER and MEGA7, a differentially expressed MYB transcription factor related to the flavonoid biosynthesis pathway was screened and named *NtMYB9*. The sequence and expression pattern of the *NtMYB9* gene were analyzed, and its function in the flavonoid biosynthesis pathway of the corona was analyzed by subcellular localization and transient transformation, with a view to providing a theoretical basis for conducting research on the flavonoid metabolic regulatory network and flower color innovation in the Chinese Narcissus.

## 2. Materials and Methods

### 2.1. Plant Materials

Chinese Narcissus 'Jinzhanyintai' plants were purchased in Zhangzhou, Fujian, China and grown in an artificial climate chamber in December at 20 °C for 16 h in the light, 16 °C for 8 h in the dark, with a humidity of 70%. The corona, tepal, filament, ovary and leaf from the same period were collected from the largest flower on the inflorescence during the five periods of the flowering process (Figure 1), wrapped with tin foil, quick-frozen in liquid nitrogen for 10 min and stored in a −80 °C refrigerator for later use.

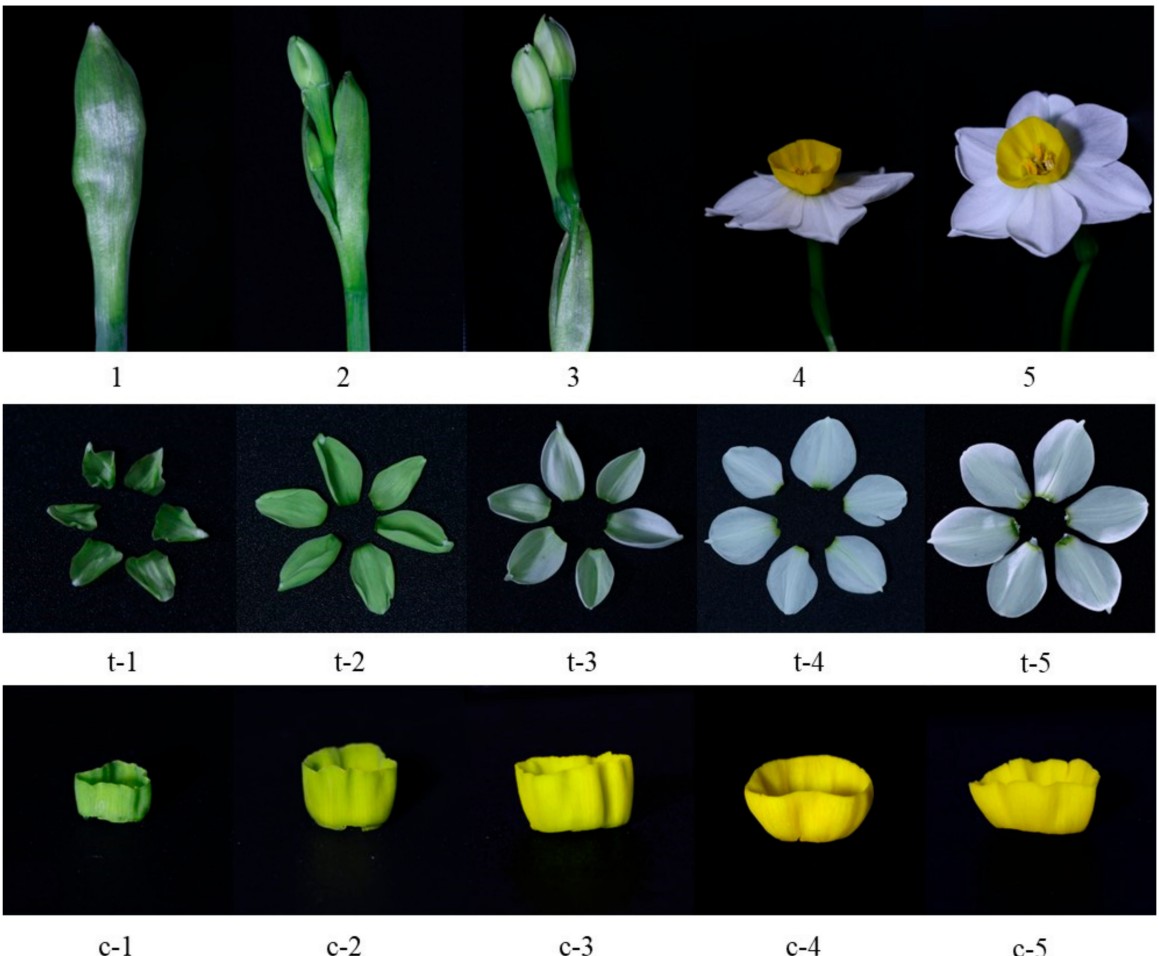

**Figure 1.** Five periods of *N. tazetta* var. *chinesis* M.Roem. 'Jinzhanyintai'. 1–5: the overall states of 'Jinzhanyintai' in five periods of the coloring process, t-1–t-5: the states of 'Jinzhanyintai' tepal in five periods of coloring process, c-1–c-5: the states of 'Jinzhanyintai' corona in five periods of the coloring process.

*Nicotiana benthamiana* (Tobacco) was grown in a light incubator at 22 °C for 22 h of light, 2 h of dark at 50–70% humidity for transient transformation assays. 'W115' *P. × hybrida* (Petunia) and 'Hangzhoumeinu' *P. aphrodite* (Phalaenopsis) were used for the transient transformation assays and grown in the artificial climate chamber at 22 °C for 14 h in the light and 20 °C for 10 h in the darkness. Petunia were given by Associate Professor Hu Huirong of Huazhong Agricultural University, Wuhan, China, and Phalaenopsis were purchased from Chaolai Spring Flower Market, Beijing, China.

*2.2. Total RNA Extraction and cDNA Synthesis*

The total RNA of the 'Jinzhanyintai' corona, tepal, filament, ovary and leaf were extracted using the Quick RNA Isolation Kit (0416-50GX, Huayueyang, Beijing, China), according to the manufacturer's instructions. The RNA concentration was tested by NanoDrop2000. And the clarity and integrity of band was checked using 1% agarose gel electrophoresis. The RNA was reverse transcribed using the Reverse Transcription Kit (A3500, Promega, Madison, WI, USA), according to the manufacturer's instructions. The RT-PCR reaction system (20 μL total volume) was as follows: 4 μL of 25 mM $MgCl_2$, 2 μL of RT 10× Buffer, 2 μL of 10 mM dNTP Mixture, 0.5 μL of Recombinant RNasin Ribonuclease Inhibitor, 0.63 μL of AMV Reverse Transcriptase, 0.5 μL of Oligo(dT)$_{15}$ Primer, 1 μg of total RNA and $ddH_2O$ up to 20 μL. The cDNA was stored in the refrigerator at −20 °C for later use.

### 2.3. Screening of NtMYB9 Gene

The basic local alignment search tool (BLAST) [22] was used to screen out homologous *NtMYB* gene sequences, based on the transcriptome data of corona at five periods in 'Jinzhanyintai' and 419 MYB protein sequences of *A. thaliana* and *Oryza sativa* from the Plant-TFDB (http://planttfdb.gao-lab.org/index.php, accessed on 12 October 2020) database [23]. HMMER (https://www.ebi.ac.uk/Tools/hmmer/search/hmmsearch, accessed on 15 October 2020) [24] was used to check the sequences of *NtMYB* for MYB conserved domains, and those without MYB domains were removed. The sequences with MYB domains were constructed in the MEGA7 software (V7.0.21) by the neighbor-joining (NJ) method [25,26] and the same sequences and the sequences that could not be clustered into one branch were removed. The *NtMYB* were compared with all the different expression genes (DEGs) in the corona transcriptome, and a MYB transcription factor gene related to flavonoid biosynthesis was screened and named *NtMYB9*.

### 2.4. Fluorescence Quantification of NtMYB9 in 'Jinzhanyintai'

Quantitative reverse-transcription PCR assays (qRT-PCR) were carried out to detect the expression level of *NtMYB9* gene in different tissues and five periods of the corona coloring process in 'Jinzhanyintai' (Figure 1). Based on the sequence of *NtMYB9*, qRT-PCR-specific primers with a single common PCR gel band were designed using SnapGene software (V2.3.2) (Table 1). The *Actin* gene (GenBank accession number JN204912.1), reported by Wang Guiqing, was used as the internal reference gene [27]. The qRT-PCR assays were performed using TB Green *Premix Ex Taq* II (RR820A, Takara, Kusatsu, Japan), following the manufacturer's recommendations. The PCR reaction system (10 μL total volume) contained 5 μL of TB Green *Premix Ex Taq*, 1 μL of cDNA template, 0.4 μL of (F + R) primer and 3.6 μL of ddH$_2$O, and three biological replicates were performed. The PCR program was carried out with 95 °C pre-denaturation for 30 s, 40 cycles of 95 °C for 5 s and 62 °C for 30 s, the melting curve was 60 °C to 95 °C and the temperature was increased by 1 °C every 15 s. When the peak of the melting curve appeared single and smooth above 85 °C, the experimental results were reliable. Then the relative expression level of the *NtMYB9* gene at different tissues and corona coloring processes in 'Jinzhanyintai' were calculated by the $2^{-\Delta\Delta Ct}$ method [28], with the expression level in the filament and in the first period of corona being the control group.

**Table 1.** The primer sequences used for PCR.

| Primer Name | Primer Sequence (5′→3′) | Description |
|---|---|---|
| *NtMYB9*-F | GCTCTAGAATGGGTAGGCACTCTTGTTGCT | PCR amplification |
| *NtMYB9*-R | TCCCCCCGGGAATCTGTCCAAACACTGAAGAAATCTTCTG | |
| q-*NtMYB9*-F | TCCCCTCTCACCCAATTGAATTC | qRT-PCR |
| q-*NtMYB9*-R | CACGGCAACATTTCACCACC | |
| q-*Actin*-F | TGCCCAGAAGTGCTATTCCAG | qRT-PCR |
| q-*Actin*-R | GTTGACCCACCACTAAGAACAATG | |
| P1300-LV-*NtMYB9*-F | GAGAACACGGAGCTCGGTACCATGGGTAGGCACTCTTGTTG | Construction of expression vector |
| P1300-LV-*NtMYB9*-R | TCCTCCTCCTCCTCCGGATCCAATCTGTCCAAACACTGAAG | |

### 2.5. Cloning of NtMYB9 Gene

Primers of the *NtMYB9* gene were designed using SnapGene software (V2.3.2) (Table 1). The target full-length gene was amplified using the LA Taq kit (iRR002A, TaKaRa) with corona cDNA as the template. PCR products were detected by 1% agarose gel electrophoresis, and the target bands were recycled using a Agarose Gel DNA Extraction Kit (DP103-03, Tiangen, Beijing, China). Then, the target sequence was ligated to the pMD19-T vector at room temperature by the pMD<sup>TM</sup>19-T Vector Cloning Kit (6013, TaKaRa) and the ligation

products were transformed into a DH5α *Escherichia coli* competent cell. The correct plasmids solution was sent to the company Anshengda, Beijing, China, for sequencing after colony PCR validation and plasmids extraction (DP103, Tiangen).

### 2.6. Bioinformatics Analysis of NtMYB9

The MYB protein sequences of the flavonoid synthesis pathways from other species were obtained from National Center for Biotechnology Information (NCBI: https://www.ncbi.nlm.nih.gov/, accessed on 17 May 2021) public databases. The multiple sequences alignment was performed utilizing the amino acid sequences of NtMYB9 and other R2R3-MYB proteins in *Citrus sinensis*, *Diospyros kaki*, *F. × ananassa*, *M. domestica*, *V. vinifera*, *Zea mays*, *P. × hybrida* and *A. thaliana* through DNAMAN software (V6.0.3.99). The phylogenetic tree was constructed with the NtMYB9 protein sequence and MYB protein sequences in 13 species using MEGA7 software through the neighbor-joining method (NJ) and 1000 bootstrap replicates. The NtMYB9 protein sequences was submitted to MEME (http://meme-suite.org/tools/meme, accessed on 21 May 2021) for motif analysis along with the MYB protein sequences from other species [29], with parameters set to an expected motif site distribution of 0 or 1, the number 8, and a width of 5 to 50.

### 2.7. Construction of Plant Overexpression Vectors

Restriction endonucleases (New England Biolabs, Inc., Ipswich, MA, USA) KpnI and BamHI were used for double digestion of the P1300-LV-GFP (35S::-GFP) plasmid (Shanghai Weidi Biotechnology Co., Ltd., Shanghai, China) to obtain single-stranded vector. Based on the digestion site of the P1300-LV-GFP vector and the CDS sequence of *NtMYB9*, seamless cloning primers were designed by SnapGene software (V2.3.2) (Table 1). Seamless cloning was performed by KOD DNA polymerase (KOD-401, TOYOBO) using the pMD19-T vector ligated with the full-length cDNA of *NtMYB9* as a template. The DNA products were recycled and ligated with the single-stranded vector plasmids of P1300-LV-GFP to construct the plant overexpression vectors, P1300-LV-*NtMYB9*-GFP (35S::*NtMYB9*-GFP), using the 2×EasyGeno Assembly Mix Kit (VI201-02, Tiangen). *NtMYB9* was expressed under the control of the 35S promoter. The 10 μL reconstitution system contained 5 μL of 2×EasyGeno Assembly Mix, 2.5 μL of the single-stranded vector plasmids and 2.5 μL of PCR fragment. The reconstitution reaction system was put in a 50 °C water-bath for 30 min, then transformed into a DH5α *E. coli* competent cell and placed in an incubator at 37 °C for 16 h. Single colonies were picked, identified by colony PCR, and sent to the company Anshengda, Beijing, China, for sequencing. The plasmids of the correctly sequenced colonies were extracted and stored in a refrigerator at 20 °C.

### 2.8. Subcellular Localization Assay of NtMYB9 Proteins in Tobacco

The plant overexpression vector plasmid, P1300-LV-*NtMYB9*-GFP, was transferred into GV3101 *Agrobacterium tumefaciens* competent cells (Shanghai Weidi Biotechnology Co., Ltd., Shanghai, China) by heat stimulation. The single colonies were cultured for 60 h and identified by colony PCR. Positive single colonies were picked and placed into 20 mL LB liquid medium containing 50 mg/L kanamycin and 25 mg/L rifampicin and incubated at 28 °C with shaking at 200 rpm until the $OD_{600}$ value was about 0.5. The bacteria were collected by centrifugation at 28 °C and 5000 rpm for 10 minutes and resuspended in infection solution (10 mM MES-KOH, 10 mM $MgCl_2$, 200 μM AS) until $OD_{600} = 0.5$ at room temperature for 2–3 h. Infection solution was injected into the back of tobacco leaves using a de-needled 1 mL sterile syringe. Transferred to a light incubator for 2–4 d after 1 d of dark incubation, after which the infected tobacco leaves were cut and stained with the DPAI solution (C0060, Solarbio) for 10 min at room temperature, rinsed 2–3 times with PBS solution. The fluorescence signal was observed using a Zeiss Axio Imager M2 microscope (Zeiss, Germany); GFP fluorescence was excited at 488 nm and DAPI fluorescence was excited at 365 nm. The plant expression vector plasmid P1300-LV-GFP was used as a positive control and treated as P1300-LV-*NtMYB9*-GFP.

*2.9. Fluorescence Quantification of Flavonoid-Related Enzyme Genes in Tobacco*

Three tobacco plants in good growth condition were selected. Their leaves were injected with infection solution with P1300-LV-*NtMYB9*-GFP. While the other three non-infected tobacco plants were selected as blank controls grown in the same conditions as the infected tobacco plants. All tobacco plants were incubated in the dark for 1 d and then transferred to a light incubator. The expression of the *NtMYB9* gene was viewed by the GFP fluorescence signal using a Zeiss Axio Imager M2 microscope. Tobacco leaves with the strongest GFP fluorescence signals and non-infected tobacco leaves from the same period were collected and immediately frozen in liquid nitrogen. RNA was extracted from the above tobacco leaf samples and reverse transcribed into cDNA.

The expression levels of *PAL* (phenylalanine ammonia-lyase), *4CL* (4-coumarate CoA ligase), *CHS* (chalcone synthase), *CHI* (chalcone isomerase), *F3′H* (flavonoid 3′-hydroxylase), *F3H* (flavonoid 3-hydrox-ylase), *FLS* (flavonol synthase), *DFR* (dihydroflavonol 4-reductase), *ANS* (anthocyanidin sythase), *UFGT* (glucose-flavonoid 3-o-glucosyltransferase), *LAR* (leucoanthocyanidin reductase) and *ANR* (anthocyanidin reductase) of the flavonoid synthesis pathway in tobacco were detected by qRT-PCR, and the primers of enzyme genes were designed with reference to Guiqing Wang (Table 2) [30].

**Table 2.** The primer sequences used for qRT-PCR.

| Primer Name | Primer Sequence (5′→3′) | Description |
|:---:|:---:|:---:|
| *PAL*-F | AACCAACAGTCAGGGGAATG | qRT-PCR |
| *PAL*-R | TTGGGCATCGAGAGTTCCAG | |
| *4CL*-F | TCATTGACGAGGATGACGAG | qRT-PCR |
| *4CL*-R | TGGGATGGTTGAGAAGAAGG | |
| *CHS*-F | GTACAACTAGTGGTGTAGACA | qRT-PCR |
| *CHS*-R | CCAACTTCACGAAGGTGAC | |
| *CHI*-F | GAAATCCTCCGATCCAGTGA | qRT-PCR |
| *CHI*-R | CAACGTTGACAACATCAGGC | |
| *F3′H*-F | TCCAAGAATACTGGCCCAAG | qRT-PCR |
| *F3′H*-R | CTCACAACTCTCGGATGCAA | |
| *F3H*-F | ACAGGGTGAAGTGGTCCAAG | qRT-PCR |
| *F3H*-R | CCTTGGTTAAGGCCTCCTTC | |
| *FLS*-F | GTCCACAACGTTGCATGGTG | qRT-PCR |
| *FLS*-R | CACAACTTCTCGCAGCCTC | |
| *DFR*-F | AACCAACAGTCAGGGGAATG | qRT-PCR |
| *DFR*-R | TTGGGCATCGAGAGTTCCAG | |
| *ANS*-F | TGGCGTTGAAGCTCATACTG | qRT-PCR |
| *ANS*-R | GGAATTAGGCACACACTTTGC | |
| *UFGT*-F | CAATGTTTGGGATGGTGTCA | qRT-PCR |
| *UFGT*-R | TTCCTCCTCTGCCTCTTTCA | |
| *LAR*-F | TCAAGGTCCTTTACGCCATC | qRT-PCR |
| *LAR*-R | ACGAACCTGCTTCTCTTTGG | |
| *ANR*-F | CATTTGACTTTCCCAAACGC | qRT-PCR |
| *ANR*-R | ATTGGGCTTTTGAGTTGTGC | |
| *ACTIN*-F | AATGATCGGAATGGAAGCTG | qRT-PCR |
| *ACTIN*-R | TGGTACCACCACTGAGGACA | |

*2.10. Transient Transformation Assay of NtMYB9 Proteins in Petunia and Phalaenopsis*

Positive single colonies were shaken in 20 mL LB liquid medium containing 50 mg/L kanamycin and 25 mg/L rifampicin at 28 °C and 200 rpm until the $OD_{600}$ value was about 0.6, centrifuged at 4 °C and 5000 rpm for 10 min to collect the bacteria, which were the GV3101 *A. tumefaciens* competent cells, in which the plant overexpression vector plasmid P1300-LV-NtMYB9-GFP was transferred. The bacteria were resuspended in the infection solution (4.43 g MS powder was added to 1 L ddH2O, sterilized at 121 °C for 25 min and 200 mmol/L AS for 500 μL was added before use) at room temperature for 2–3 h until $OD_{600}$ = 0.6, generating the suspension [31]. Then, the suspension was slowly injected into newly opened petals of Petunia and fully opened petals and sepals of Phalaenopsis using a de-needled 1 mL sterile syringe. Petunia and Phalaenopsis were incubated in the dark for 1 d, then transferred to the artificial climate chamber and continuously observed after transient transformation.

## 3. Results and Analysis

*3.1. Determination of the Periods of Corona Coloring Process*

The corona coloring process of the 'Jinzhanyintai' was divided into five periods (Figure 1). 'Jinzhanyintai' were cultivated with hydroponics until the scape was drawn and the states of involucre and inflorescence during the flowering process were observed. The color of the corona showed a transition from green to yellow as the flowers of 'Jinzhanyintai' opened. The transformation of the flower color was recorded with reference to the fifth edition of the Royal Horticultural Society color chart. The corona coloring process was divided into five periods. In the first period, before the rupture of the involucre of the inflorescence, the corona was green, with the color code GREEN GROUP 143-B; in the second period, after the rupture of the involucre, the inflorescence was fully extracted and the florets were exposed, the corona was yellow-green, with the color code YELLOW–GREEN GROUP 144-A; in the third period, the tepal was slightly expanded, the corona was yellow-green, with the color code YELLOW–GREEN GROUP N144-A; in the fourth period, the flowers were fully opened, the tepal was fully expanded, the corona was yellow, with color code YELLOW GROUP 13-A; in the fifth period, 4 d, after the flowers were fully opened, the tepal and corona color were the same as the fourth period, and the corona was yellow, with color code YELLOW GROUP 13-A.

*3.2. Screening for a Transcription Factor Involved in Corona Coloring*

Previous studies have shown that *NtMYBs* is related to the color-forming of the yellow corona. A *NtMYB* gene was a differentially expressed gene (DEG) in the third-generation and second-generation transcriptome database of the corona coloring process (unpublished). Then, the *NtMYB* gene was obtained using the MYB protein sequences of *A. thaliana* and *O. sativa* as query sequences and identified by HEMMER and MEGA7 as a member of the MYB gene family, named *NtMYB9* with the GenBank accession number OM925904.

*3.3. Analysis of Tissue Expression Specificity and Cloning of NtMYB9*

To investigate the function of *NtMYB9* in the coloration process of 'Jinzhanyintai', the expression pattern of *NtMYB9* in *N. tazetta* var. *chinensis* M.Roem. was quantified by qRT-PCR. The result showed that the expression level of the *NtMYB9* gene was significantly different in different tissues of 'Jinzhanyintai' (Figure 2A) and was significantly higher in the corona than in other tissues, followed by in the tepal, and almost not expressed in the filaments. *NTMYB9* was four times more abundantly expressed in the corona than in the tepal. The expression of *NtMYB9* was also detected in five periods of the corona coloration process. The result revealed that *NtMYB9* was expressed in all five periods (Figure 2B), with the highest expression level in the fourth period of complete coloration, a low expression level in the third and fifth periods and almost no expression in the first and second periods. This result was consistent with the expression level of *NtMYB9* in the

transcriptome database. Then, the full-length sequence of the *NtMYB9* gene was obtained by cloning.

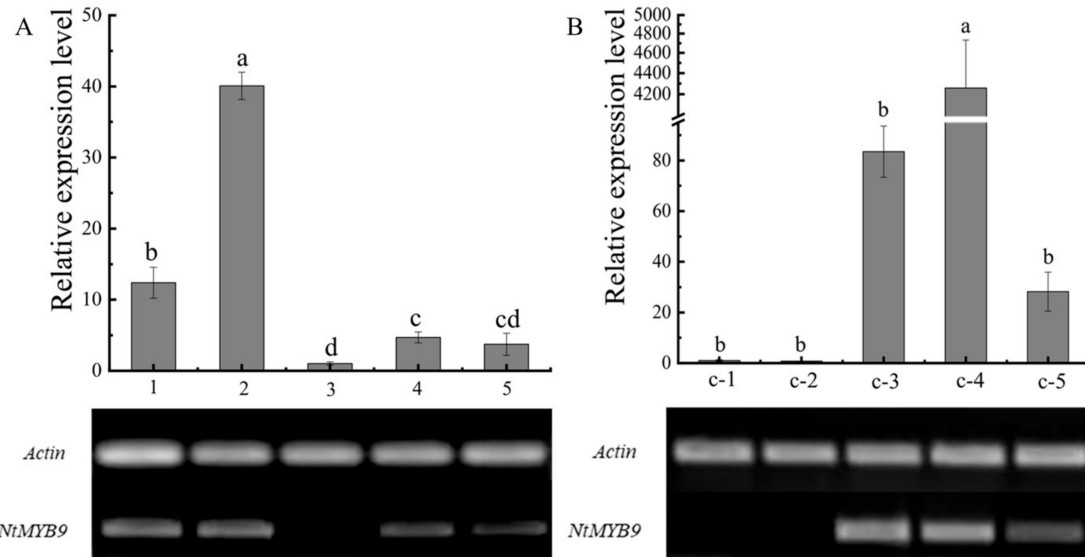

**Figure 2.** The expression level of the *NtMYB9* gene in different tissues and different periods of the corona coloring process in 'Jinzhanyintai'. (**A**) The expression level of the *NtMYB9* gene in five different tissues, 1 to 5 are tepal, corona, filament, ovary and leaf, respectively. (**B**) The expression level of the *NtMYB9* gene in the five periods of corona coloring process. c-1 to c-5 were the first period to the fifth period of the corona coloring process, respectively. Different small letters indicated significant difference at $p < 0.05$ level by Duncan test.

### 3.4. Bioinformatics Analysis of NtMYB9 Protein

The bioinformatics analysis of *NtMYB9* was analyzed. The results revealed that the open reading frame (ORF) of the *NtMYB9* gene was 1008 bp, which encoded 336 amino acids. A sequence analysis revealed that NtMYB9 protein contained the R2 and R3 MYB DNA-binding domains, indicating that NtMYB9 protein was a R2R3-MYB transcription factor (Figure 3). The phylogenetic tree, constructed by MEGA7 software, revealed that 26 MYB sequences from 13 species were divided into two groups. The activators of the flavonoid biosynthesis pathway were clustered into one group, including three branches of the Anthocyanin pathway, Proanthocyanin pathway and flavonol pathway, in which NtMYB9 protein and flavonol activators were on one branch, such as AtMYB11 and AtMYB111 from *A. thaliana* [7], MdMYB22 [32] in *M. domestica* and CsMYBF1 [33] in *C. sinensis*. The inhibitors of flavonoid synthesis pathway were clustered into one group, including MYB4-like and FaMYB1-like branches (Figure 4A). The conserved motifs of the 26 MYB proteins sequences from 13 species were analyzed, and a total of 8 conserved motif were identified (Figure 4B). Motif 1 was R3-MYB, which was presented in all branches. Motif 2 was the R2-MYB and was highly conserved. Motif 7 was presented only in the branches of Anthocyanin pathway and Proanthocyanidin pathway. Motif 5 and Motif 8 were specific to MYB4-like branch and had the function of MYB4-like proteins.

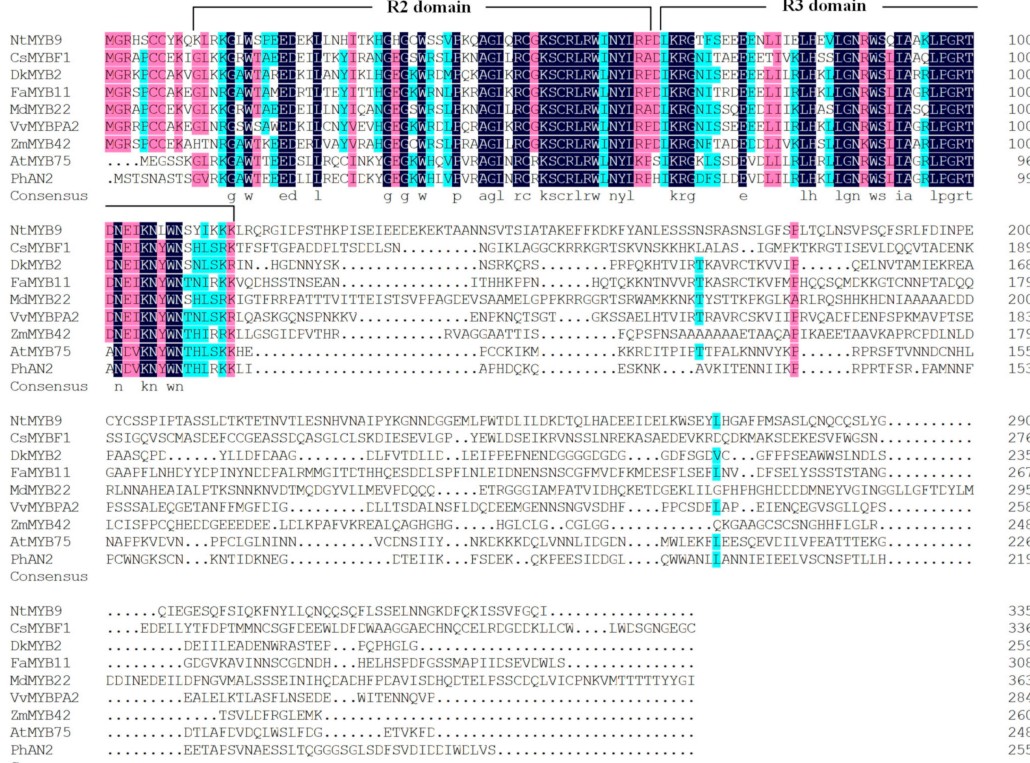

**Figure 3.** Multiple sequences alignment of the NtMYB9 and other MYB proteins in *N. tazetta* var. *chinensis* M.Roem. (Nt), *C. sinensis* (Cs), *D. kaki* (Dk), *F. × ananassa* (Fa), *M. domestica* (Md), *Vitis vinifera* (Vv), *Z. mays* (Zm), *A. thaliana* (At) and *P. × hybrida* (Ph). The R2 and R3 domains shown refer to two MYB DNA-binding domains of the MYB proteins. Blue–black indicated amino acid sequence similarity of 100%, pink indicated amino acid sequence similarity of greater than or equal to 75% and blue indicated amino acid sequence similarity of greater than or equal to 50%.

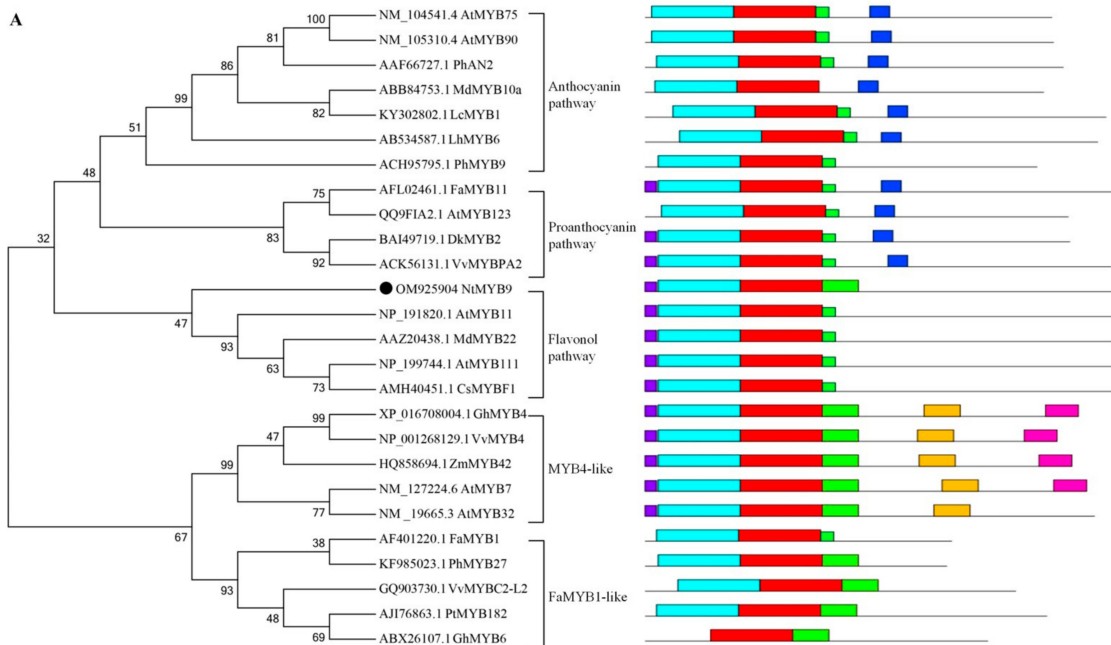

**Figure 4.** *Cont.*

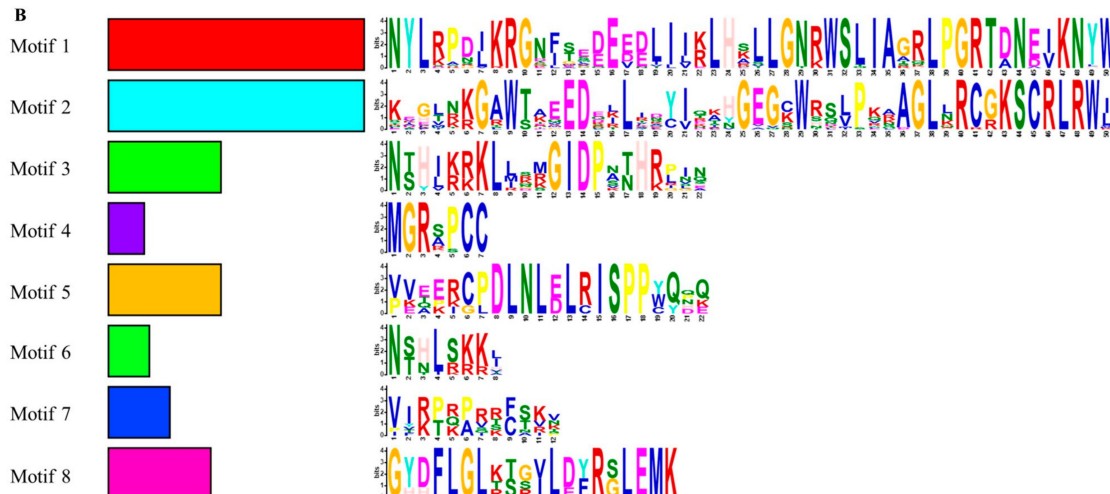

**Figure 4.** Phylogenetic and conserved motif analysis of MYB proteins. (**A**) Phylogenetic tree and conserved motif analysis of *NtMYB9* and other MYB proteins (including GenBank accession numbers and MYB protein names) known to regulate the flavonoid biosynthesis pathway in *A. thaliana* (At), *P. × hybrida* (Ph), *M. domestica* (Md), *Litchi chinensis* (Lc), *Lilium hybrid* (Lh), *V. vinifera* (Vv), *Phalaenopsis hybrid* (Ph), *F. × ananassa* (Fa), *D. kaki* (Dk), *N. tazetta* var. *chinensis* M.Roem. (Nt), *C. sinensis* (Cs), *Gossypium hirsutum* (Gh), *Z. mays* (Zm) and *Populus tremula × Populus tremuloides* (Pt), the black dot was the NtMYB9 protein. (**B**) Sequence identification of the predicted eight motifs of the MYB proteins.

### 3.5. Subcellular Expression of NtMYB9 in Tobacco

In order to explore the subcellular localization of the NtMYB9 protein, the *NtMYB9* gene was constructed into the P1300-LV-GFP vector and expressed as a fusion with the GFP gene. The fluorescence signals of the *NtMYB9* protein were observed under microscope. It was found that the green fluorescence signals excited by GFP were localized in the nucleus of epidermal cells in tobacco and overlapped with the blue fluorescence signals excited by DAPI, while the control group was observed with green fluorescence signals in both the cell membrane and nucleus, and the green fluorescence signals in the nucleus overlapped with the blue fluorescence signals (Figure 5). It indicated that the *NtMYB9* gene mainly functions in the nucleus as a transcription factor.

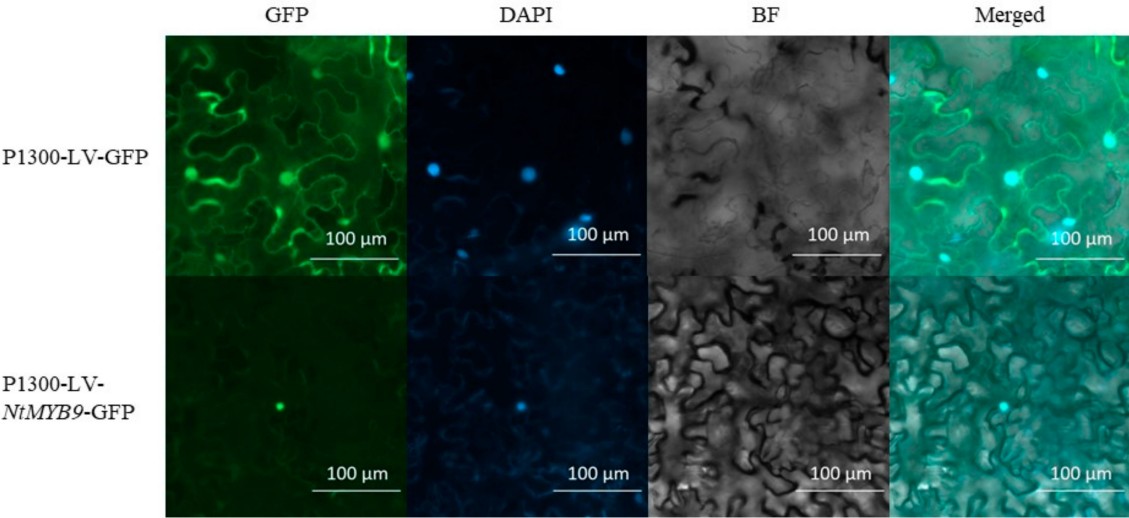

**Figure 5.** Subcellular localization analysis of P1300-LV-*NtMYB9*-GFP fusion proteins in epidermal cells of tobacco. The scale bar is 100 μm.

*3.6. Effect of Transient Expression of NtMYB9 on Expression of Flavonoid-Related Genes in Tobacco*

To investigate the function of the *NtMYB9* gene, *NtMYB9* gene was overexpressed in tobacco leaves. The expression level of the enzyme genes related to flavonoid biosynthesis in common and transgenic tobacco plants was analyzed by qRT-PCR assays. This revealed that the expression levels of *PAL* and *4CL* in transgenic tobacco leaves were significantly up-regulated and the expression levels of *CHI*, *F3′H*, *FLS*, *DFR* and *UFGT* were up-regulated, while the expression levels of *CHS*, *F3H*, *ANS*, *LAR* and *ANR* were not significantly changed compared with common tobacco (Figure 6).

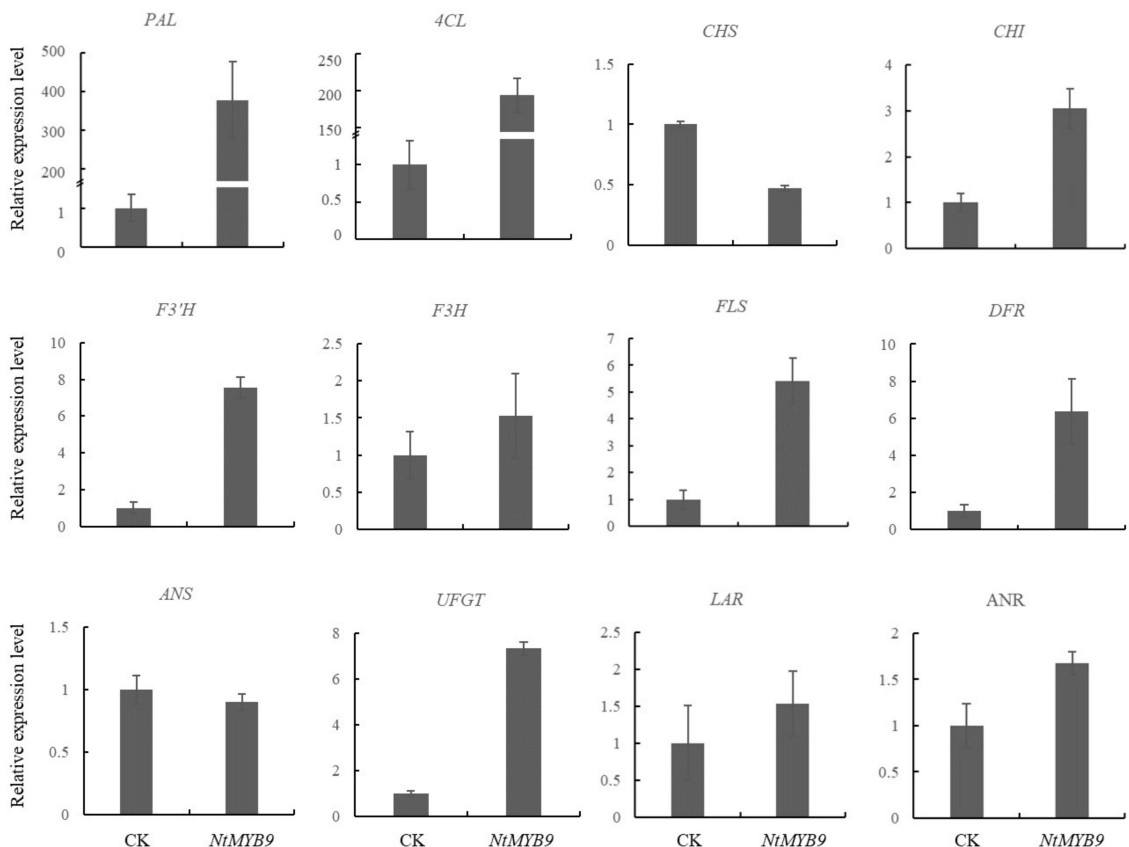

**Figure 6.** The expression level of enzyme genes of flavonoid biosynthesis pathway in tobacco leaves.

*3.7. Effect of Transient Expression of NtMYB9 in Petunia and Phalaenopsis on Coloring*

To further investigate the function of *NtMYB9*, P1300-LV-*NtMYB9*-GFP fusion overexpression vectors were transformed into 'W115' Petunia and 'Hangzhoumeinv' Phalaenopsis. The overexpression vectors of P1300-LV-*NtMYB9*-GFP and P1300-LV-GFP transformed into 'W115' Petunia and the petals of transgenic Petunia became significantly more yellow and were similar to the base of petals compared with those transferred into P1300-LV-GFP in color (Figure 7A). The flower colors of the transgenic Phalaenopsis became significantly more yellow compared with the control group (Figure 7B). The results indicated that the *NtMYB9* gene had a role in promoting the synthesis of yellow metabolites in the petals and sepals of Phalaenopsis.

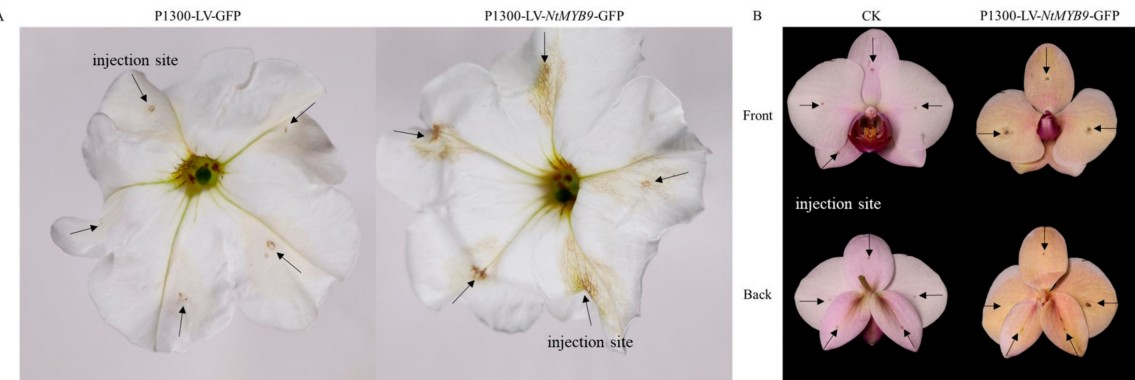

**Figure 7.** Functional verification of the *NtMYB9* gene in the transient expression system of Petunia and Phalaenopsis. (**A**) Transient expression of the NtMYB9 protein in petals of 'W115' Petunia. Overexpression of the *NtMYB9* gene caused yellowing around the injection sites of Petunia petals, and the discoloration areas were similar in color to the base of the petals. (**B**) Transient expression of the NtMYB9 protein in petals and sepals of 'Hangzhoumeinv' Phalaenopsis. Overexpression of the *NtMYB9* gene caused yellowing around the injection sites of Phalaenopsis petals and sepals. The black arrows indicate the injection sites.

## 4. Discussion

*N. tazetta* var. *chinensis* M.Roem. have a high economic value as annual flowers. The pigment components in the yellow corona of Chinese Narcissus mainly include flavonol and flavonoid metabolites in the flavonoid synthesis pathway. With the growth and development of ornamental plants, the pigment content in flowers changes continuously. The color of the corona shows a transition from green to yellow with the opening of flowers in 'Jinzhanyintai'. In this study, the coloration process of the corona was divided into five periods. Due to the spatiotemporal specificity of gene expression in plants, it was found using fluorescence quantitative assays that the *NtMYB9* gene had the highest expression in the corona of 'Jinzhanyintai'. The expression level of the *NtMYB9* gene was continuously up-regulated during the coloration process, with a sudden increase in the fourth period when the corona becames completely yellow and a significant decrease in the fifth period. Therefore, it is speculated that the *NtMYB9* gene mainly functions during the fourth period of the corona coloration process and plays a role in the synthesis of metabolites related to the flavonoid synthesis pathway.

MYB transcription factors are widely involved in the flavonoid biosynthesis pathway of plants [34,35]. The results of subcellular localization assays showed that the NtMYB9 protein was localized in the nucleus of epidermal cells in tobacco leaves, which indicated that the *NtMYB9* gene had the general characteristics of transcription factors. Sequence analysis revealed that NtMYB9 protein was the R2R3-MYB transcription factor with the motifs of R2-MYB and R3-MYB and was highly homologous to other R2R3-MYB plants, such as MdMYB22 in *M. domestica*. Phylogenetic tree analysis revealed that the NtMYB9 protein was clustered with other flavonol activators, such as AtMYB11 and AtMYB111 in *A. thaliana* and CsMYBF1 in *C. sinensis*, and they could control the synthesis of flavonols by activating the expression of related genes (*CHS*, *CHI*, *F3H*, *FLS*, etc.) in the flavonoid biosynthesis pathway [36]. Therefore, it was hypothesized that the *NtMYB9* gene plays a transcription-promoting role in the flavonoid biosynthetic pathway by activating the expression of enzyme genes.

Tobacco and Petunia are commonly used for the verification of flower color genes. Their transformation systems have matured, and good experimental results are obtained [30,37]. Transient transformation is widely used to verify the function of plant flower color genes and is a technique with a short experimental period and the ability to obtain transient high expression level of the target genes [38]. The flavonoid biosynthesis pathway is known to have several branches, such as the flavonol pathway, anthocyanin pathway and

proanthocyanidin pathway [39]. In the flavonoid biosynthesis pathway, Phenylalanine is synthesized as Dihydroflavonol by *PAL*, *C4H*, *4CL*, *CHS*, *CHI*, *F3′H*, *F3H* and *F3′5′H* [40]. Dihydroflavonol is produced as a flavonol by *FLS*; Dihydroflavonol can also be synthesized as Leucoanthocyanidin by *DFR*; Leucoanthocyanidin is synthesized as Anthocyanin by *ANS*; Anthocyanin is modified to form stable and colorful Arachidoside by *UFGT* and *MT*; and Leucoanthocyanidin and Anthocyanin are synthesized as Proanthocyanidin by *LAR* and *ANR*, respectively [41]. It was found that the color of transgenic tobacco leaves had no significant change from transient transformation. Furthermore, fluorescence quantitative PCR assays revealed that the expression levels of seven flavonoid-related genes, *PAL*, *4CL*, *CHI*, *F3′H*, *FLS*, *DFR* and *UFGT*, were significantly increased in tobacco leaves with the overexpression of the *NtMYB9* gene. It indicated that the *NtMYB9* gene as a transcriptional activator could promote the expression level of most enzyme genes in flavonoid biosynthesis pathway of tobacco. The expression levels of *FLS* and *DFR* were both up-regulated in the transgenic tobacco, so it was hypothesized that the content of flavonols and leucoanthocyanidins were increased in the leaves of transgenic tobacco. As the expression level of *ANS*, *LAR* and *ANR* had no significant change, the leucoanthocyanidins accumulated by *DFR* were not converted into anthocyanins and proanthocyanidins. The color of flavonols is usually yellowish, so the accumulation of flavonols content did not lead to significant changes in tobacco leaf color.

Further validation through the transient transformation of 'W115' Petunia and 'Hangzhoumeinv' Phalaenopsis showed clearly that the overexpression of the *NtMYB9* gene turned the flowers of Petunia and Phalaenopsis yellow. As for the tissue specificity of the gene expression in the plants, the petals in petunia are white and the petal bases are yellow, so the genes controlling the synthesis of yellow metabolites are expressed in the petal bases. The overexpression of the *NtMYB9* gene into petunia petals showed that the petal discoloration sites were similar to the color of the petal bases. It was speculated that the *NtMYB9* gene might promote the expression level of flavonoid-related genes in 'W115' petunia petals to produce yellow metabolites to make the petals yellow. The yellow metabolites might be flavonols. However, this speculation needs to be further verified. Both *FLS* and *DFR* act on Dihydroflavonol in the flavonoid biosynthesis pathway, so there is substrate competition between the flavonol pathway and the anthocyanin pathway [42]. The main pigment component in the petals and sepals of Phalaenopsis is anthocyanins [43], and flavonoids and flavonols are auxiliary pigments of arachidosides [44]. The flavonoid metabolites in the corona of Chinese Narcissus are mainly flavonols, and since there is no ANS gene in Chinese Narcissus, the flavonoid synthesis pathway cannot flow toward the anthocyanin synthesis pathway [27]. All of these led to the speculation that the overexpression of the *NtMYB9* gene allowed the substrate that synthesizes arachidoside to be used for the synthesis of flavonols, causing Phalaenopsis flowers to turn yellow, but this speculation needs further research.

## 5. Conclusions

In this study, a R2R3-MYB transcription factor gene with R2 and R3 conserved motifs and the highest expression and specificity in the corona of *N. tazetta* var. *chinesis* M.Roem. was identified by bioinformatics, cloned and named *NtMYB9*. *NtMYB9* played a role in the corona coloring process. Furthermore, the *NtMYB9* gene could up-regulate the expression level of related genes and promote the synthesis of metabolites in the flavonoid biosynthesis pathway.

**Author Contributions:** Conceptualization, K.F., Y.M., Z.J. and T.H.; methodology, K.F., Y.M. and X.H.; software, K.F. and Y.C.; validation, W.Z., Y.D. and Y.C.; data curation, K.F. and X.H.; writing—original draft preparation, K.F., Y.M. and M.X.; writing—review and editing, K.F., Y.M. and T.H. All authors have read and agreed to the published version of the manuscript.

**Funding:** This research was supported by ICBR Fundamental Research Funds Grant (NO. 1632020021 and NO. 1632021018).

**Institutional Review Board Statement:** Not applicable.

**Informed Consent Statement:** Not applicable.

**Data Availability Statement:** All data in this study can be found in the manuscript.

**Conflicts of Interest:** The authors declare no conflict of interest.

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
