# Peer review of "Cloning and Functional Analysis of NtMYB9 in ‘Jinzhanyintai’ of Narcissus tazetta var. chinensis"

_horticulturae, doi:10.3390/horticulturae8060528_

Round 1
Reviewer 1 Report
Maybe it was interesting (for the ones who do not know the flower market in China) to have a cultivar (if Jinzhanyintai is a cultivar) description and why this one is so relevant to study.
And also the complete latin name of the species (N. tazetta var. chinensis M.Roem.) for a better identification.
Author Response
Hello, this is my response.
Please see the attachment.

Reviewer 2 Report
The manuscript “Cloning and Functional Analysis of NtMYB9 in ‘Jinzhanyintai’ of Narcissus tazetta var. chinensis” by Fan et al. describes the characterization, cloning, transient expression, and functional characterization of a myb transcription factor from Narcissus tazetta var. chinensis. The goal of this study was to identify a regulator of yellow color development in the corona of N. tazetta var. chinensis, and flower petals in general. The spesies is of great ornamental value, and understanding yellow color development in flowers is of interest beyond this species.
The myb transcription factor NtMYB9 was found by computational analysis of transcriptomic data. The screening itself appears well done and is interesting, and should be explained briefly in abstract and introduction, and in more detail in the results (it is currently only explained in the materials and methods section).
Some of the computational characterization of NtMYB9 appears random. Why showing results that this transcription factor has no transmembrane motifs nor signal peptides? One would not expect those for a transcription factor. It may be fine to mention this in a sentence when stating the nuclear location, but certainly no need to show these results in a figure. And what is the purpose of the secondary structure prediction? It doesn’t seem to add any important information.
The expression level of NtMYB9 gene in N. tazetta var. chinensis was assessed by qRT-PCR, and was highest in the corona after it just turned yellow, supporting a possible role of NtMYB9 in the color development.
Transient expression of NtMYB9 in tobacco revealed up-regulation of genes related to flavonoid biosynthesis pathway. Transient expression of NtMYB9 in Petunia and Phalaenopsis showed significant yellow color development.
Taken together, these results suggest that NtMYB9 gene is involved in the positive regulation of flavonoid biosynthesis and yellow color development in flowers, a trait of great economic importance. The results look convincing, though I would not have expected that a transcription factor from a monocot flower correctly regulates genes in distant eudicots, such as tobacco and petunia; a point that should be discussed.
The language needs revisions for clarity. Also, the order of the results could be improved; I made suggestions below
Additional Points
ABSTRACT
“A MYB gene was screened from the transcriptome of the corona in this study”
mention briefly how this screening was done
INTRODUCTION
“Chinese Narcissus have fewer variety resources because they lack the key gene ANS for anthocyanin synthesis”
What are variety resources? Do you mean “less color variety”?
Separate into 2 sentences:
“Compared with other flowers, Chinese Narcissus have fewer variety resources because they lack the key gene ANS for anthocyanin synthesis, and have a single flower color, mainly yellow and white, among which the ‘Jinzhanyintai’ has six white tepals and a yellow wineglass-shaped corona.”
Shorten run-away sentences; all this is one sentence, but could be 3-4!
“The MBW (MYB-bHLH-WD40) complex formed by the three is also jointly involved in the expression of related enzyme genes in the flavonoid synthesis pathway [6], of which MYB is one of the transcription factor families with the largest number of studies and the most extensive roles in ornamental plants, and most MYB transcription factors belong to R2R3-MYB [7], which directly regulates the transcription and expression of structural genes by binding to their promoters [8], thereby activating or repressing the synthesis of flavonoid metabolites [9], which in turn affects the production and accumulation of phytochrome [10].”
“In recent years, R2R3-MYB transcription factors have been screened and cloned”
Screened for, how?
“In this study, based on the transcriptomic data of the corona coloring process”
Vague, how did you identify this specific myb gene?
The screening is later explained in the Materials and Methods section, but a brief explanation would be helpful here
MATERIALS AND METHODS
“…(Phalaenopsis) were used to for the transient expression assays”
“Real-time fluorescence quantification PCR assays (qRT-PCR)”
qRT-PCR is spelled out incorrectly.
Correct: qRT-PCR (quantitative reverse-transcription PCR)
The q in qRT-PCR stands for quantitative
RT in qRT-PCR stands for reverse transcription (not for real-time, that’s already indicated by q)
Give more detail on the qRT-PCR: primer design, controls, melting curve analysis, standard curve to calculate amplification efficiency…!
“different periods of corona coloring process…”
refer to figure 1, which explains the 5 periods of coloring
“PCR fragments were recycled”
What does that mean?
“The Actin gene, reported by Wang Guiqing, was used as the internal reference gene…”
Was this reference gene determined as stabile, e.g. by geNORM, under the same conditions as used here?
“…rpm for 10 min to collect the bacteria”
state what bacteria you are referring to
“The bacteria was resuspended in infection solution”
The bacteria were resuspended in infection solution
Throughout the text: keep in mind that “bacteria” is the plural form
“Positive single colonies were shaken into 20 mL LB”
Positive single colonies were shaken in 20 mL LB
RESULTS
Suggestions to re-organize the results sections (just suggestions, I trust the authors to improve the order of results as they see fit):
- Determination of the Periods of the Corona Coloring Process
[this can go first, if important for explaining the screening approach)
- Screening for a Transcription factor involved in Corona Coloring
- Computational Analysis of the NtMYB9 protein [this may be combined with point 2 if it is considered part of the screening, and should include MSA and phylogenetic tree, but not figure 2, which should be removed)]
- Analysis of Tissue Expression Specificity of NtMYB9 [this point confirms that NtMYB9 is indeed a promising candidate, so I would place it before the cloning]
- Cloning of NtMYB9 and Subcellular Expression in Tobacco
- Effect of Transient Expression of NtMYB9 on Expression of Flavonoid-related Genes in Tobacco
- Effect of Transient Expression of NtMYB9 Protein in Petunia and Phalaenopsis on Coloring
“The corona coloring process of the ‘Jinzhanyintai’ was determined into five periods”
The corona coloring process of the ‘Jinzhanyintai’ was divided into five periods
“Observing the state of ‘Jinzhanyintai’ during the flowering process (including the states of involucre, inflorescence and scape) after hydroponics to protrude the flower buds.”
I don’t understand this sentence
“The samples of c-1 to c-5 were subjected to third-generation and second-generation transcriptome sequencing to obtain transcriptome data of corona coloring process.”
The transcriptomics are not subject of this study (or else would have to be explained in much more detail). Are authors referring to another study? If so, cite, or a “(unpublished)”.
Shorten any portions of the computational analysis that doesn’t reveal interesting information; I would take out figure 2.
FIGURES
Figure. 7 The expression level of structural genes of flavonoid biosynthesis pathway in tobacco leaves.
Throughout the text, don’t call these “structural;” genes. These genes encode enzymes within pathways, while “structural” implies non-enzymatic proteins.
Figure 8A.
Mark the injection sites, e.g. with errors; I don’t know where to look for yellow color, both flowers look similar to me.
DISCUSSION
Discuss the fact that this transcription factor apparently regulates genes in distant species, which would indicate that the promoter motifs must be very conserved. Are there other examples of transcription factor from a monocot species that work in eudicots (or vice versa) to support this finding?
CONCLUSION
“In this study, a R2R3-MYB transcription factor gene with R2 and R3 conserved motifs, named NtMYB9, was cloned from N. tazetta var. chinesis and identified by bioinformatics.”
Again, consider the correct order of points: first the gene was identified, then it was cloned
Author Response

(The authors gave the same response as above.)

Reviewer 3 Report
Narcissus tazetta var. chinensis (Chinese Narcissus) is a popular flower in Asia. Its corona shows a transition from green to yellow, which is a good material to study flower color forming. The authors focused on NtMYB9, an MYB family gene in Chinese Narcissus, which was found from transcriptome different expression gene (DEGs) results. The bioinformatic analysis confirmed it contained conserved MYB domains. The qRT-PCR showed that NtMYB9 was significantly expressed in period 4 in corona. Overexpression of NtMYB9 in tobacco leaf suggested it may up-regulate flavonoid biosynthesis-related genes. Overexpression of NtMYB9 in Petunia and Phalaenopsis showed yellowing in petals or sepals. These suggested the NtMYB9 may be involved in color forming in Nt corona.
The paper was well organized, the methods seemed fine, and the results were reasonable. I recommend accepting this article for Horticulturae. I just want to ask small questions.
1. I can't find the NtMYB9 sequence in Genebank of accession number OM925904, is that correct?
2. The authors said the NtMYB9 comes from the screening of the available third-generation and second-generation transcriptome database of the corona at five periods in 'Jinzhanyintai' in section 3.2, is there has reference, database link, or SRA link? And why is the MYB named NtMYB9, was it the ninth named in Nt, or has a relationship with a known MYB9? I didn't find a relationship in the phylogenetic tree.
3. Why not do overexpression or knockdown of NtMYB9 in Narcissus tazetta directly?
Minor:
section 2.4, Zea mays must be in italics.
Thanks
Author Response

(The authors gave the same response as above.)

Reviewer 4 Report
The work of Fan, Ma and colleagues characterizes the role of ntMYB9 in the colour forming in Chinese Narcissus.
They analyse the expression profile in the flower during its formation, and evaluate the effect of its overexpression both in tobacco leaves and in other plants. They observe its implication in the synthesis of flavonoids and importantly its effect in colour modification in the transient transformation of Petunia and Phalaenopsis.
Overall the work is logically presented and the experiments proposed represent a rigorous approach for the understanding of role of the MYB gene in analysis in the colour profile of an horticulturally important plant like the Chinese Narcissus .
I consider the paper presents interesting findings in the flavonoids biosynthesis, with an economic relevance.
Author Response
Thanks very much for the carefully review and kind comments.
We wish you all the best!